# Prevalence of body mass index categories among adults living alone in China: Observational study

**Zhixuan Li**[1☯], **Mengmeng Yan**[2☯]*, **Yingying Liu**[3]

**1** Urban Vocational College of Sichuan, Chengdu, China, **2** School of healthcare and technology, Chengdu Neusoft University, Chengdu, China, **3** Department of Health Management & Institute of Health Management, Sichuan Provincial People's Hospital, University of Electronic Science and Technology of China, Chengdu, China

☯ These authors contributed equally to this work.
* changc99@outlook.com

## Abstract

### Background

Adults living alone represent a growing population group in China. Understanding the prevalence of body mass index (BMI) categories and their associations with demographic and lifestyle factors among this group is essential for informing targeted interventions and public health policies.

### Methods

In this population-based cross-sectional study, we used individual-level data from the 2011–2021 China General Social Survey. Main outcomes were prevalence of BMI categories adjusted for gender and age, using logistic regression and model-predicted marginal prevalence to estimate BMI categories prevalence.

### Results

We analyzed 9,077 single-living Chinese adult participants. The primary-adjusted prevalence of BMI categories varied across different genders and age groups. Underweight was more prevalent in females (12.73%; 95% CI: 12.31% - 13.14%) than in males (7.54%; 95% CI: 7.19% - 7.88%), while overweight and obesity were higher in males. Primary-adjusted underweight prevalence was highest among the 18–24 years age group (22.09%; 95% CI: 20.17% - 24.01%) and decreased with age. Primary-adjusted overweight prevalence increased with age, peaking in the 45–54 years age group (41.94%; 95% CI: 40.96% - 42.93%). Primary-adjusted obesity prevalence exhibited a fluctuating pattern across age groups, with the highest prevalence observed in the 45–54 years age group (9.81%; 95% CI: 9.19% - 10.44%).

### Conclusion

Our findings reveal significant associations between BMI categories and demographic and lifestyle factors among adults living alone in China. These results can inform targeted

**Data Availability Statement:** All data files are available from the CGSS database (http://cgss.ruc.edu.cn/English/Home.htm).

**Funding:** The funders had no role in study design, data collection and analysis, decision to publish, or preparation of the manuscript.

**Competing interests:** The authors have declared that no competing interests exist.

interventions and public health policies aimed at promoting healthy weight management and addressing the unique health challenges faced by single-living individuals in China.

## Introduction

The prevalence of different body mass index (BMI) categories, including underweight, overweight, and obesity, has become a significant public health concern worldwide. These BMI categories are associated with various health risks, such as cardiovascular diseases, diabetes, and certain types of cancer [1–4]. Living alone refers to residing by oneself, and it is common in single-person households. In recent years, there has been a growing interest in understanding the health challenges faced by adults living alone, as this population group has been increasing globally [5,6]. For example, China experienced a 32.3% surge between 2003 and 2021 [7]. As of 2020, the estimated number of single-person households in China was 125 million [7]. Previous studies have reported higher risks of unhealthy weight and poor dietary habits among individuals living alone [8–11]. Besides those, the number of adults living alone has been increasing in China, partly due to the growing aging population and the preference for independent living among younger generations [12].

However, there is limited research on the prevalence of BMI categories and their associations with demographic and lifestyle factors among adults living alone in China. Such information can help inform targeted interventions and public health policies aimed at promoting healthy weight management and addressing the unique health challenges faced by single-living individuals in China.

In this study, we utilized the China General Social Survey (CGSS) [13] data to estimate the prevalence of BMI categories among adults living alone in China between 2011 and 2021. We also examined the associations between demographic and lifestyle factors and BMI categories. Our findings contribute to a better understanding of the prevalence of BMI categories among adults living alone in China and their associations with demographic and lifestyle factors, which can inform targeted interventions and public health policies.

## Methods

### 1. Data source

We obtained the CGSS data from 2011–2013,2015,2017–2018, and 2021 to estimate BMI categories prevalence among adults LA in China. CGSS is a comprehensive national survey project jointly conducted by the Department of Sociology at Renmin University of China and the Institute of Sociology at Chinese Academy of Social Sciences. Each round of research uses multi-stage stratified design and data quality controls to ensure the reliability and accuracy of the research results [13]. The sample covers different people and social classes in cities and rural areas across all provinces, autonomous regions, and municipalities in China, thereby having a certain degree of representativeness. The final CGSS data set included 9077 adults. (Exclusion criteria and respondent characteristics are provided in S1 File).

### 2. Study variables

Participants' BMI categories were classified based on their self-reported weight and height using BMI cutoffs modified for Asian adults (<18.5 kg/m$^2$, 18.5 to 22.9 kg/m$^2$, 23 to 27.4 kg/m$^2$, ≥27.5 kg/m$^2$) [14]. Participants were categorized into "live alone" (adults living without

children) based on self-reported living arrangements (specific survey questions are listed in S1 Table in S1 File).

The CGSS collected demographic and socioeconomic information, including age, gender, ethnic, educational level, physical activity, health insurance, and family annual income (Specific survey questions for the variables are listed in S1 Table in S1 File).

### 3. Statistical analysis

The direct method was used to calculate the crude prevalence of BMI categories among adults living alone. Logistic regression was used with BMI categories as the outcome variable to estimate the adjusted prevalence of BMI categories. The model-predicted marginal prevalence provided marginal standardized estimates while adjusting for primary variables and covariables. Then we evaluated linear trends of adjusted overweight and obesity prevalence over time were on survey years (modeled as a continuous independent variable). In the sensitivity analysis, we used standard BMI categorizations in CGSS. In addition, we employed Probit regression analysis instead of logistic regression analysis to reevaluate the prevalence of overweight and obesity. We also examined the urban-rural differences in BMI among the solitary population.

Statistical significance was nonoverlapping 95% CIs. And all p-values are less than 0.001, unless otherwise specified. Analyses were done using Stata, version 17.0 (USA) and accounted for the respondent sampling weights. As specified in their statistical analysis guidelines [13]. The authors had no access to information that could identify individual participants during or after data collection.

## Results

### 1. Participants characteristics among adults living alone

Between 2011 and 2021, CGSS included 9077 LA participants, with a balanced gender distribution, higher prevalence of older and less-educated individuals, predominantly Han ethnicity, and a greater proportion of low-income earners (Table 1). The crude prevalence of underweight, normal weight, overweight, and obesity among adults living alone was found to be 13.2% (12.2%, 14.2%), 49.8% (48.4%, 51.3%), 29.8% (28.5%, 31.1%), and 7.2% (6.5%, 7.9%) respectively.

### 2. Adjusted BMI categories prevalence

We found that the primary-adjusted prevalence of BMI categories varied by gender and age (Fig 1). For gender, primary-adjusted prevalence of BMI categories across different genders revealed that underweight was higher in females (12.73%; 95% CI: 12.31% - 13.14%) compared to males (7.54%; 95% CI: 7.19% - 7.88%). In contrast, overweight and obesity were more prevalent in males (overweight: 36.76%; 95% CI: 36.15% - 37.36%; obesity: 9.85%; 95% CI: 9.47% - 10.23%) than in females (overweight: 30.92%; 95% CI: 30.38% - 31.47%; obesity: 7.48%; 95% CI: 7.16% - 7.80%). The prevalence of normal weight was relatively similar between both genders (males: 45.86%; 95% CI: 45.22% - 46.50%; females: 48.87%; 95% CI: 48.27% - 49.48%). Regarding age groups, the primary-adjusted prevalence of BMI categories displayed considerable variation. The primary-adjusted prevalence of underweight, normal weight, overweight, and obesity across different age groups showed distinct patterns. Primary-adjusted underweight prevalence was highest among the 18–24 years age group (22.09%; 95% CI: 20.17% - 24.01%) and decreased with age. Primary-adjusted overweight prevalence increased with age, peaking in the 45–54 years age group (41.94%; 95% CI: 40.96% - 42.93%). Primary-adjusted

**Table 1. Participants characteristics from CGSS.**

| Characteristics | LA (n = 9077) |
|---|---|
| BMI categories | |
| Underweight | 13.2(12.2,14.2) |
| Normal | 49.8(48.4,51.3) |
| Overweight | 29.8(28.5,31.1) |
| Obesity | 7.2(6.5,7.9) |
| Female | 48.9(47.5,50.3) |
| Age groups | |
| 18-24y | 7.6(6.9,8.4) |
| 25-34y | 12.3(11.4,13.3) |
| 35-44y | 8.7(7.9,9.5) |
| 45-54y | 17.2(16.1,18.3) |
| 55-64y | 18.2(17.1,19.3) |
| >64y | 36.1(34.7,37.5) |
| Education | |
| Primary school | 45(43.6,46.5) |
| Up to high school | 37.6(36.2,39) |
| College graduate | 17.4(16.3,18.4) |
| Han | 92.9(92.1,93.6) |
| Health insurance | 88.1(87.2,89) |
| Annual income | |
| <¥20000 | 40.4(39,41.9) |
| ¥20000-¥40000 | 24.6(23.4,25.9) |
| ¥40000-¥75000 | 19.5(18.4,20.7) |
| >¥75000 | 15.4(14.5,16.5) |
| Physical activity | 50(48.6,51.5) |

LA = living alone.

Education below primary school includes no education, private school and primary school. Up to high school include junior high schools, vocational high schools, regular high schools, technical secondary schools and technical schools. College graduate includes junior college degree or bachelor degree, graduate degree or above.

Annual income is classified by the sample quartile, considering weight.

Data presented are weighted percent (95% CI) indicating population distribution of each characteristic, unless otherwise specified. All the P-values are less than 0.001.

obesity prevalence exhibited a fluctuating pattern across age groups, with the highest prevalence observed in the 45–54 years age group (9.81; 95% CI: 9.19% - 10.44%).

The covariates-adjusted prevalence (Fig 1) of underweight was still higher in females compared to males, with females showing a prevalence of 15.75% (95% CI: 14.26% - 17.24%) and males at 10.56% (95% CI: 9.28% - 11.85%). The covariates-adjusted prevalence of overweight exhibited a slight increase for both genders, with males at 31.47% (95% CI: 29.61% - 33.32%) and females at 28.10% (95% CI: 26.30% - 29.90%). Lastly, the covariates-adjusted prevalence of obesity showed a small increase for females to 7.84% (95% CI: 6.79% - 8.89%) and a slight decrease for males to 6.55% (95% CI: 5.59% - 7.50%) compared to the primary-adjusted prevalence. Regarding age groups, the covariates-adjusted prevalence of underweight generally increased for the younger age groups and decreased for the older age groups. The covariates-adjusted prevalence of overweight showed an increase for most age groups, while the adjusted prevalence of obesity exhibited a mixed pattern, with some age groups experiencing an increase and others a decrease.

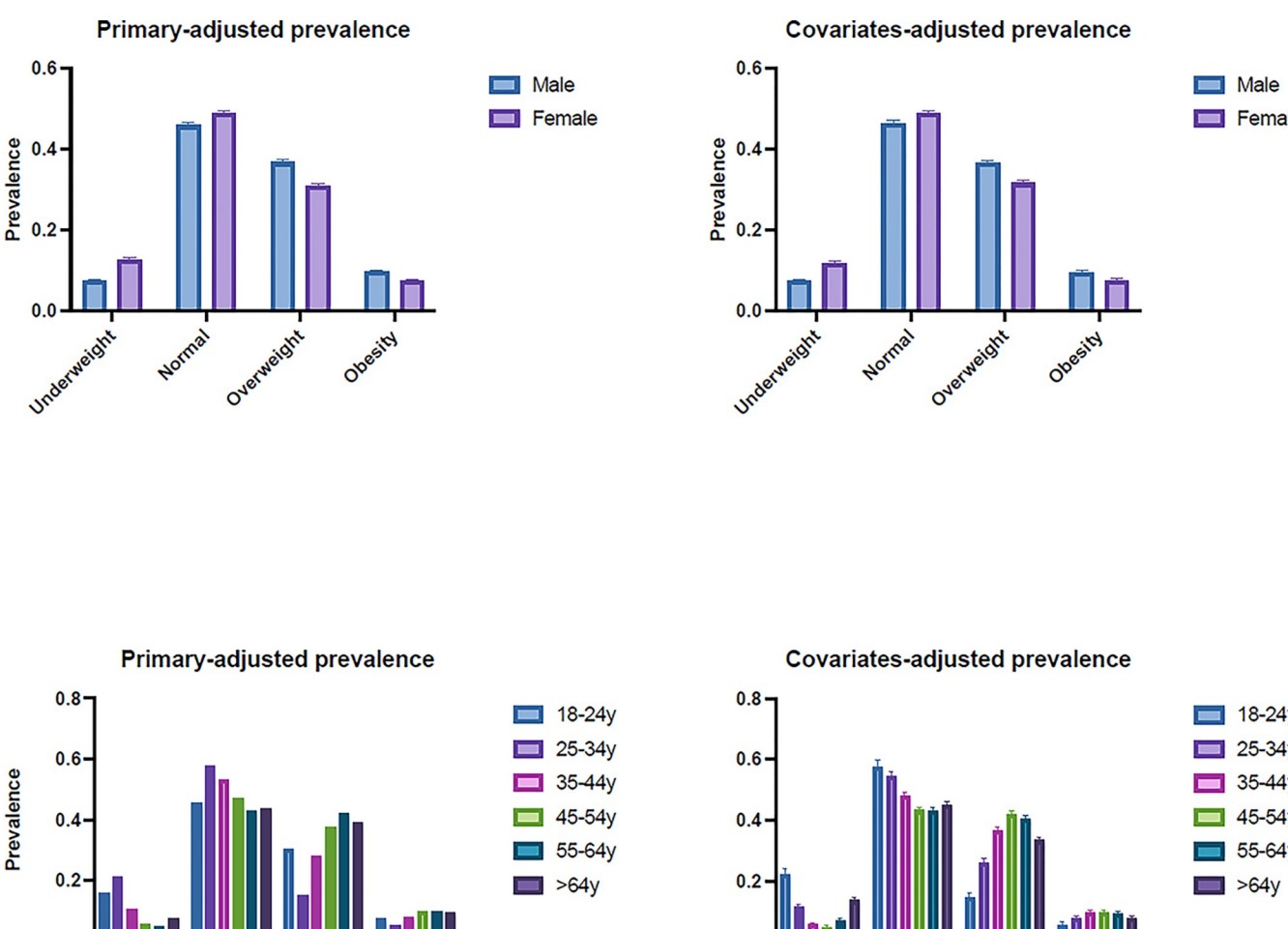

**Fig 1. Adjusted prevalence of BMI categories among Chinese adults.** Primary-adjusted prevalence refers to adjusted for sex and age, covariates-adjusted prevalence refers to adjusted for sex, age, demographic, and lifestyle factors.

S1 Fig in S1 File illustrates the interaction between age and gender, with the obesity prevalence among women slightly higher than men after the age of 35, while overweight prevalence in men is slightly higher than women across all age groups.

## 3. Relevant variables

We calculated the odds ratios (ORs) for various factors associated with BMI categories. The results revealed significant associations between BMI categories and gender, age, ethnicity, physical activity, health insurance, income, and education levels (Fig 2A–2C). For the overweight category, factors such as age, income, and education levels showed positive associations, while females (OR: 0.94, 95% CI: 0.82–1.08) and individuals with higher physical activity (OR: 1.11, 95% CI: 0.95–1.30) had lower odds of being overweight. Ethnic minority status and health insurance showed mixed associations with overweight prevalence. For the underweight category, females (OR: 1.62, 95% CI: 1.33–1.97) and ethnic minorities (OR: 1.26, 95% CI: 0.89–1.78) had higher odds of being underweight, while age, physical activity (OR: 0.66, 95%

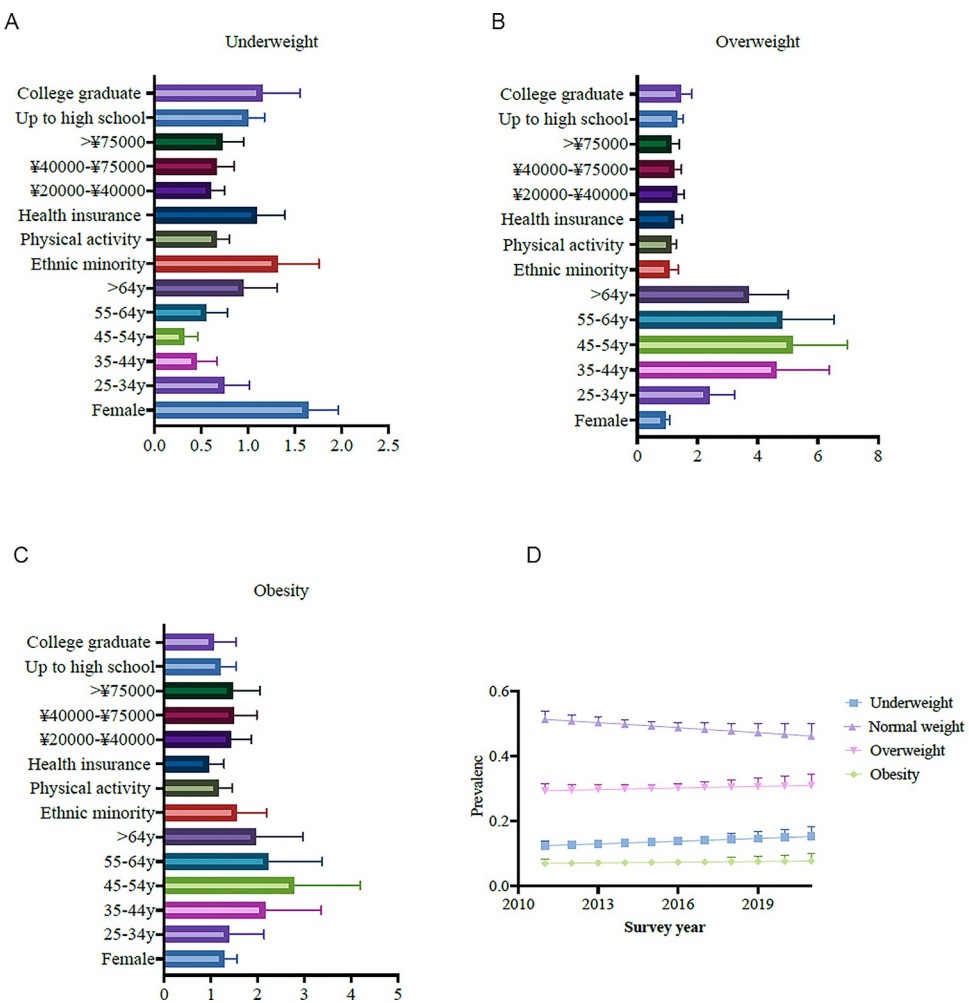

**Fig 2. Association between covariates and the prevalence of BMI categories.** Figures A-C represent the odds ratio for covariates and prevalence of BMI categories, while Figure D represents the time trend of BMI category prevalence after adjusting for covariates.

CI: 0.53–0.81), health insurance (OR: 1.07, 95% CI: 0.81–1.41), income, and education levels showed negative associations. The odds of being underweight generally decreased with increasing age. For the obesity category, age, income, and ethnic minority status (OR: 1.48, 95% CI: 0.98–2.23) showed positive associations with obesity prevalence, while females (OR: 1.26, 95% CI: 1.02–1.57), physical activity (OR: 1.14, 95% CI: 0.88–1.47), health insurance (OR: 0.91, 95% CI: 0.64–1.30), and college education (OR: 0.99, 95% CI: 0.63–1.57) had lower odds of being obese. The odds of being obese generally increased with age, peaking in the middle-age groups.

We also investigated the prevalence of four BMI categories (underweight, normal weight, overweight, and obesity) over the years 2011 to 2021. The results show a slight rise in the covariates-adjusted prevalence prevalence of underweight, overweight, and obesity categories (Fig 2D).

## 4 Sensitivity analysis

After adjusting the classification threshold of BMI, we found generally comparable results to the main analysis. S2 Table in S1 File presents the results of the Probit regression model

estimates for the prevalence of overweight and obesity, which are roughly consistent with the main results. S2 Fig in S1 File displays the estimated proportions and their 95% confidence intervals of solitary individuals in urban and rural areas across four BMI categories. In urban areas, the proportion of individuals with low body weight is 12.14% (95% CI: 9.35% - 14.94%), which is higher than the 8.44% (95% CI: 5.37% - 11.51%) observed in rural areas. However, in the normal body weight category, the proportion in rural areas is 67.86% (95% CI: 62.39% - 73.34%), slightly higher than the 63.26% (95% CI: 58.97% - 67.55%) in urban areas. The proportion of overweight individuals is similar in both urban (21.53%, 95% CI: 17.88% - 25.17%) and rural (21.05%, 95% CI: 16.33% - 25.77%) areas. Likewise, the proportion of obesity is also similar in urban (3.07%, 95% CI: 1.52% - 4.62%) and rural (2.65%, 95% CI: 0.69% - 4.61%) areas. Overall, this reflects significant differences between urban and rural areas in the distribution of low body weight and normal body weight, while the differences in overweight and obesity are relatively small.

## Discussion

This study aimed to investigate the prevalence of BMI categories among adults living alone in China and examine the associations between demographic and lifestyle factors and BMI categories. Our findings demonstrated significant differences in the associations for underweight, overweight, and obesity categories, as well as changes in the prevalence of each BMI category over time. These results contribute to a better understanding of the unique health challenges faced by the solitary-living population in China and can inform targeted interventions and public health policies.

The higher prevalence of underweight, overweight, and obesity among adults living alone compared to the general population is consistent with previous studies that have reported higher risks of unhealthy weight among individuals living alone [15–17]. The observed differences in the associations between BMI categories and demographic and lifestyle factors, such as gender, age, ethnicity, physical activity, health insurance, income, and education levels, highlight the importance of considering these factors when designing interventions and policies targeting adults living alone in China.

Our findings also revealed that the prevalence of overweight and obesity was higher among males, older age groups, and urban residents. This is in line with previous research showing that males and older individuals are more likely to be overweight or obese [18–20]. Additionally, urban residents in China may have greater exposure to obesogenic environments, such as increased availability of unhealthy food options and reduced opportunities for physical activity, which can contribute to higher rates of overweight and obesity [21,22].

Out results showed a slight rise in the covariates-adjusted prevalence of underweight, overweight, and obesity categories. These findings are consistent with previous research that has reported an increasing trend in the prevalence of general and abdominal obesity among Chinese adults over the past few years [22–25]. The rise in obesity prevalence among Chinese urban residents may be attributed to various factors, such as unhealthy dietary habits, lack of physical activity, lifestyle changes, and socioeconomic development [23]. Additionally, research has found gender differences in the association between food away-from-home consumption and body weight outcomes among Chinese adults, which may impact the prevalence of different BMI categories [26,27].

The changes in the prevalence of BMI categories over the years 2011 to 2021 underscore the need for continuous monitoring and evaluation of weight-related trends within the solitary-living population in China. This can help identify emerging patterns and inform the development of timely and effective interventions to address the specific needs of this population group.

Regarding the prevalence and trends in obesity, our study reveals the evolving obesity rates in China. These findings are consistent with previous research. Jia et al. demonstrated a temporal trend in childhood obesity prevalence in China, along with demographic and geographical disparities [28]. Hu et al. (2017) investigated the prevalence of overweight, obesity, abdominal obesity, and related risk factors in southern China [29].

In the context of trends in living alone and their health impacts, our study findings indicate an association between living alone and health. This observation aligns with existing research. For instance, Zhong et al. reported a relationship between loneliness and cognitive function in older adults based on a long-term study in China [30]. Additionally, Cacioppo et al. highlighted the potential adverse health effects of loneliness and proposed underlying mechanisms [31].

Furthermore, our study explores the associations between living alone, obesity, dietary habits, gender, and age. Our findings are in line with the JAGES study in Japan, which revealed that living alone and eating habits jointly influence unhealthy dietary behaviors, obesity, and underweight in older Japanese adults [32]. This underscores the connection between living alone and dietary patterns and body weight. Moreover, Huang et al. emphasized the psychological health challenges faced by women during the COVID-19 pandemic, offering insights into the importance of gender in the health context [33]. Finally, Steyn and Mchiza delved into the relationship between obesity and the nutrition transition in sub-Saharan Africa [34].

In other countries, particularly in Japan [32] and South Korea [17], there is a correlation between living alone and overweight or obesity among the elderly population. In the Japanese study, it was found that 16% of males and 28% of females sometimes or regularly dine alone, with a significant proportion of them living alone. These individuals who dine alone and live alone tend to have a higher risk of obesity, especially among males, as compared to those who dine with others. Additionally, there may be a connection between dining alone and unhealthy eating behaviors, as well as underweight issues among males. In the South Korean study, it was observed that obesity levels among elderly individuals living alone were associated with factors at individual, social, and environmental levels. Different interventions may be needed based on the severity of obesity, with social interaction and environmental factors playing crucial roles in both obesity and overweight.

When discussing BMI, it is important to consider several factors that may influence BMI values. These factors include psychosocial status, genetic factors, and physical health, all of which can help explain why different individuals have different BMI values. Firstly, psychosocial status plays a significant role in the formation of BMI. An individual's emotional state, ability to cope with stress, and social support system can influence their dietary habits and lifestyle choices. Mental health issues and chronic stress have been linked to unhealthy eating behaviors and weight gain [31]. Additionally, social support systems and personality traits may also impact an individual's dietary choices and weight management. Secondly, genetic factors are crucial contributors to BMI variations [35,36]. Research has shown that an individual's family history and genetic background may influence their weight. Certain genetic variations can lead to differences in metabolic rates, affecting weight. However, genetic factors are just one of the many factors influencing BMI, with dietary and exercise habits remaining equally essential. Lastly, physical health status is closely related to BMI [17,27]. Chronic diseases, metabolic disorders, and medication treatments can all affect an individual's weight. For instance, some medications may lead to weight gain, while certain chronic illnesses can influence metabolic rates, impacting BMI.

Moreover, the relationship between diet type and daily calorie intake with BMI has not been extensively investigated in our study. However, existing research has demonstrated a

strong correlation between calorie intake and weight gain [37]. High-calorie diets are typically associated with weight gain, while low-calorie diets aid in weight control.

Our study has some limitations. The cross-sectional nature of the CGSS data limits our ability to establish causal relationships between the variables. And the self-reported nature of the data may be subject to recall and reporting biases. Firstly, self-reported data may be influenced by individual subjectivity and memory biases, potentially leading to inaccuracies or incomplete information. Secondly, social desirability and response biases could result in participants providing answers they perceive as socially acceptable or favorable rather than the true situation. Additionally, sampling bias may exist as some individuals may be unwilling to participate in self-report studies, or self-selection bias could be present, introducing potential distortions. Despite these limitations, our study provides valuable insights into the prevalence of BMI categories among adults living alone in China and their associations with demographic and lifestyle factors.

In conclusion, our study highlights the importance of understanding the prevalence of BMI categories and their associations with demographic and lifestyle factors among the solitary-living population in China. The findings can inform targeted interventions and public health policies aimed at promoting healthy weight management and addressing the unique health challenges faced by single-living individuals in China. Further research is needed to explore the underlying causes of the observed trends and to evaluate the effectiveness of interventions targeting this population group.

## Supporting information

**S1 File. Supporting information.**
(DOCX)

**S2 File. STROBE checklist.**
(DOCX)

**S3 File.**
(PDF)

**S4 File.**
(PDF)

## Author Contributions

**Conceptualization:** Mengmeng Yan.

**Data curation:** Zhixuan Li, Mengmeng Yan.

**Formal analysis:** Zhixuan Li, Mengmeng Yan.

**Investigation:** Zhixuan Li, Mengmeng Yan, Yingying Liu.

**Methodology:** Zhixuan Li, Mengmeng Yan, Yingying Liu.

**Project administration:** Yingying Liu.

**Resources:** Yingying Liu.

**Software:** Zhixuan Li, Yingying Liu.

**Validation:** Yingying Liu.

**Visualization:** Zhixuan Li, Mengmeng Yan, Yingying Liu.

**Writing – original draft:** Mengmeng Yan, Yingying Liu.

**Writing – review & editing:** Zhixuan Li, Mengmeng Yan, Yingying Liu.

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
