## [Decision Letter · Decision Letter 0]

14 Sep 2023

PONE-D-23-14813Prevalence of Body Mass Index Categories Among Adults living alone in China: Observational StudyPLOS ONE

Dear Dr. Yan,

Thank you for submitting your manuscript to PLOS ONE. After careful consideration, we feel that it has merit but does not fully meet PLOS ONE’s publication criteria as it currently stands. Therefore, we invite you to submit a revised version of the manuscript that addresses the points raised during the review process.

We look forward to receiving your revised manuscript.

Kind regards,

Zhuo Chen, Ph.D.

Academic Editor

PLOS ONE

Journal Requirements:

4. Please include a copy of Table 1 which you refer to in your text on page 3. 

Additional Editor Comments:

The paper reads well in general. I suggest that the authors look into different modeling strategies, e.g., Yen et al. (2009) that used switching regression with known regime.

There are some minor editorial/cosmetic issues. For example, LA needs to be defined first -- it appears to be living alone from the context -- and used consistently.

Yen ST, Chen Z, Eastwood DB. (2009) Lifestyle and obesity: a multi-regime switching regression analysis with an endogenous regressor. Health Services Research, 44:1345-1369.

Reviewers' comments:

Reviewer's Responses to Questions

**Comments to the Author**

1. Is the manuscript technically sound, and do the data support the conclusions?

Reviewer #1: Partly

2. Has the statistical analysis been performed appropriately and rigorously? 

Reviewer #1: Yes

3. Have the authors made all data underlying the findings in their manuscript fully available?

Reviewer #1: Yes

4. Is the manuscript presented in an intelligible fashion and written in standard English?

Reviewer #1: Yes

5. Review Comments to the Author

Reviewer #1: Overall, this is a well-conducted study examining an important public health issue - the prevalence of BMI categories and associated factors among adults living alone in China. The manuscript is technically sound, with appropriate methods and robust data analysis. The results offer valuable insights into weight status and related trends in this growing demographic group. However, there are some areas that need strengthening to enhance the clarity, quality, and impact of the work:

1. The introduction provides relevant background but contains some repetitive details. Try to synthesize and condense the background concisely to highlight gaps this study aims to address.

2. Should improve the graph quality of the results tables and figure

3. Carefully proofread the entire manuscript to improve grammar, wording, and readability issues.

4. The methods should explain the CGSS sampling scheme and representativeness more clearly. Discuss any limitations of self-reported data.

5. Include p-values alongside the 95% CIs for ORs in tables/figures.

6. Use consistent formatting for income categories in Table 1 versus Figure 2.

7. Could examine interactions between key variables like gender and age.

8. The discussion could articulate the implications and importance of the findings more clearly, especially for public health practice and policymaking. Cite the following papers to enrich the discussion and background:

On living alone trends and health impacts:

Zhong, B.L., Chen, S.L., Tu, X. and Conwell, Y., 2017. Loneliness and cognitive function in older adults: findings from the Chinese longitudinal healthy longevity survey. The Journals of Gerontology: Series B, 72(1), pp.120-128.

Cacioppo, J.T., Hawkley, L.C., Crawford, L.E., Ernst, J.M., Burleson, M.H., Kowalewski, R.B. and Malarkey, W.B., 2002. Loneliness and health: Potential mechanisms. Psychosomatic medicine, 64(3), pp.407-417.

On obesity prevalence and trends in China:

Jia, P., Xue, H., Zhang, J., & Wang, Y. (2017). Time Trend and Demographic and Geographic Disparities in Childhood Obesity Prevalence in China—Evidence from Twenty Years of Longitudinal Data. International Journal of Environmental Research and Public Health, 14. https://doi.org/10.3390/ijerph14040369.

Hu, L., Huang, X., You, C., Li, J., Hong, K., Li, P., Wu, Y., Wu, Q., Wang, Z., Gao, R., Bao, H., & Cheng, X. (2017). Prevalence of overweight, obesity, abdominal obesity and obesity-related risk factors in southern China. PLoS ONE, 12. https://doi.org/10.1371/journal.pone.0183934.

On associations between living alone, obesity, diet, gender, age:

Tani, Y., Kondo, N., Takagi, D., Saito, M., Hikichi, H., Ojima, T., & Kondo, K. (2015). Combined effects of eating alone and living alone on unhealthy dietary behaviors, obesity and underweight in older Japanese adults: Results of the JAGES. Appetite, 95, 1-8. https://doi.org/10.1016/j.appet.2015.06.005.

Huang, Chengyue, et al. "Mental toll on working women during the COVID-19 pandemic: An exploratory study using Reddit data." PloS one 18.1 (2023): e0280049.

Steyn, N., & Mchiza, Z. (2014). Obesity and the nutrition transition in Sub‐Saharan Africa. Annals of the New York Academy of Sciences, 1311. https://doi.org/10.1111/nyas.12433.

6. PLOS authors have the option to publish the peer review history of their article (what does this mean?). If published, this will include your full peer review and any attached files.

Reviewer #1: No

---

## [Author Response · Author response to Decision Letter 0]

16 Nov 2023

Response to the reviewer comments for PONE-D-23-14813, "Prevalence of Body Mass Index Categories Among Adults living alone in China: Observational Study"

I would like to thank the Editor(s) and the reviewers for their constructive comments and questions on our manuscript. I believe addressing these has helped us improve the paper substantially. Please find below a response to the specific points raised by the reviewers.

Editor Comments:

The paper reads well in general. I suggest that the authors look into different modeling strategies, e.g., Yen et al. (2009) that used switching regression with known regime.

There are some minor editorial/cosmetic issues. For example, LA needs to be defined first -- it appears to be living alone from the context -- and used consistently.

Yen ST, Chen Z, Eastwood DB. (2009) Lifestyle and obesity: a multi-regime switching regression analysis with an endogenous regressor. Health Services Research, 44:1345-1369.

Response:

Thank you for your valuable feedback. We appreciate your suggestion to explore different modeling strategies, such as the one used by Yen et al. (2009) involving switching regression with known regimes. However, due to the specific application domain, data constraints, and the inherent uncertainties surrounding the regime, we opted for an alternative modeling approach, namely Probit regression, as a substitute for logistic regression. Additionally, we have addressed the concern regarding the definition of 'LA' by providing a clear definition for it in the introduction section of the paper. The changes have been marked in red.

Response: 

Comments from the external peer reviewers

1. The introduction provides relevant background but contains some repetitive details. Try to synthesize and condense the background concisely to highlight gaps this study aims to address.

Response: Thank you for your feedback. We have revised the introduction section to condense the background information and eliminate repetitive details. The revised introduction now provides a more concise overview of the relevant background while emphasizing the specific gaps that our study aims to address. The changes have been marked in green.

2. Should improve the graph quality of the results tables and figure

Response: Thank you for your feedback. We have made the necessary improvements to enhance the quality of the results tables and figures, with particular attention to Figure 2. 

3. Carefully proofread the entire manuscript to improve grammar, wording, and readability issues.

Response: Thank you for your suggestion. We have carefully proofread the entire manuscript to address any grammar, wording, and readability issues.

4. The methods should explain the CGSS sampling scheme and representativeness more clearly. Discuss any limitations of self-reported data.

Response: Thank you for your valuable feedback. We have included detailed information about the CGSS sampling scheme and its representativeness in the supplementary materials. Additionally, we have discussed the limitations of self-reported data in the discussion section of the manuscript. The changes have been marked in green.

5. Include p-values alongside the 95% CIs for ORs in tables/figures.

Response: Thank you for your suggestion. As all the p-values are less than 0.001, we did not include them in the main body of the tables/figures. However, we have now added the p-values in methods section for clarity.

6. Use consistent formatting for income categories in Table 1 versus Figure 2.

Response: Thank you for pointing out the formatting inconsistency in the income categories between Table 1 and Figure 2. We will ensure that the formatting for income categories is made consistent across both the table and the figure in the revised version."

7. Could examine interactions between key variables like gender and age.

Response: Thank you for your suggestion. We have added the analysis of interactions between age and gender in S1 Figure.

8. The discussion could articulate the implications and importance of the findings more clearly, especially for public health practice and policymaking. Cite the following papers to enrich the discussion and background 

Response: Thank you for your input. The implications of our findings for public health and policymaking have been clarified in the discussion. The suggested papers have also been included to enrich both the background and discussion sections.

---

## [Decision Letter · Decision Letter 1]

18 Dec 2023

PONE-D-23-14813R1Prevalence of Body Mass Index Categories Among Adults living alone in China: Observational StudyPLOS ONE

Dear Dr. Yan,

Thank you for submitting your manuscript to PLOS ONE. After careful consideration, we feel that it has merit but does not fully meet PLOS ONE’s publication criteria as it currently stands. Therefore, we invite you to submit a revised version of the manuscript that addresses the points raised during the review process.

We look forward to receiving your revised manuscript.

Kind regards,

Zhuo Chen, Ph.D.

Academic Editor

PLOS ONE

Journal Requirements:

Additional Editor Comments:

Please review comments from the reviewers and address them as appropriate.

Reviewers' comments:

Reviewer's Responses to Questions

**Comments to the Author**

1. If the authors have adequately addressed your comments raised in a previous round of review and you feel that this manuscript is now acceptable for publication, you may indicate that here to bypass the “Comments to the Author” section, enter your conflict of interest statement in the “Confidential to Editor” section, and submit your "Accept" recommendation.

Reviewer #1: All comments have been addressed

Reviewer #2: All comments have been addressed

Reviewer #3: All comments have been addressed

2. Is the manuscript technically sound, and do the data support the conclusions?

Reviewer #1: Yes

Reviewer #2: Yes

Reviewer #3: Yes

3. Has the statistical analysis been performed appropriately and rigorously? 

Reviewer #1: Yes

Reviewer #2: Yes

Reviewer #3: Yes

4. Have the authors made all data underlying the findings in their manuscript fully available?

Reviewer #1: Yes

Reviewer #2: Yes

Reviewer #3: Yes

5. Is the manuscript presented in an intelligible fashion and written in standard English?

Reviewer #1: Yes

Reviewer #2: Yes

Reviewer #3: Yes

6. Review Comments to the Author

Reviewer #1: The authors have addressed majority questions raised by the Editor and reviewers. Figures still need higher resolution and legends.

Reviewer #2: I appreciate the opportunity to critically review the article entitled 'Prevalence of Body Mass Index Categories Among Adults living alone in China: Observational Study' (PONE-D-23-14813R1), which addresses an interesting and significant public health issue, defining a population that would benefit from specific prevention strategies.

I believe the article is well-written, and the results and conclusions are well-founded. The observations made by the previous reviewer were highly relevant and adequately addressed by the authors. I consider that the article can be published and will be of interest to the readers of the Journal.

I commend the researchers for their work, which I, as a public health professional, thoroughly enjoyed.

Reviewer #3: Dear Editor,

Thank you for considering me as a reviewer for this publication in your esteemed journal

PLOS ONE. I have provided my comments as follows:

• Some research suggests that other measures of body fat, such as skinfold thicknesses, bioelectrical impedance, underwater weighing, and dual energy x-ray absorption, may be more accurate than BMI. The waist circumference (sometimes divided by height) is also a simple measure of fat distribution. Of course, checking waist circumference is an easy and less expensive method that can help enrich the results of this research. Therefore, it is suggested to check and add waist circumference and its relationship with BMI.

• It is suggested to compare BMI in different regions of China (urban and rural).

• It is suggested that the results of this study be compared with the results in other countries of this region.

• • As you know, some factors such as psychosocial status, genetic factors and physical health can affect BMI. Please add explanations about these factors in the discussion.

• Has the relationship between the type of diet (daily calorie intake) and BMI been investigated? Please explain.

7. PLOS authors have the option to publish the peer review history of their article (what does this mean?). If published, this will include your full peer review and any attached files.

Reviewer #1: No

Reviewer #2: No

Reviewer #3: No

---

## [Author Response · Author response to Decision Letter 1]

21 Dec 2023

Response to the reviewer comments for PONE-D-23-14813R1, "Prevalence of Body Mass Index Categories Among Adults living alone in China: Observational Study"

I would like to thank the reviewers for their constructive comments and questions on our manuscript. I believe addressing these has helped us improve the paper substantially. Please find below a response to the specific points raised by the reviewers.

Journal Requirements:

Response: Thank you very much for your guidance. We have reviewed all the references to ensure their appropriateness and made modifications to reference number 14. We have also added red markings in the manuscript to indicate these changes, as per your request.

Comments from the external peer reviewers

Reviewer #1

The authors have addressed majority questions raised by the Editor and reviewers. Figures still need higher resolution and legends.

Response: We greatly appreciate your valuable feedback on our manuscript. Regarding the issue you mentioned regarding image resolution, we would like to clarify that the images we uploaded initially met the required resolution. However, the problem arose during the conversion of these images into PDF files, which resulted in a loss of clarity in the final PDF.

To address this concern, we have taken the necessary steps to reprocess all the images and have uploaded them in PDF format. We hope this improvement will meet your expectations and provide higher clarity and quality for the images in the final version. 

Reviewer #3

1. Some research suggests that other measures of body fat, such as skinfold thicknesses, bioelectrical impedance, underwater weighing, and dual energy x-ray absorption, may be more accurate than BMI. The waist circumference (sometimes divided by height) is also a simple measure of fat distribution. Of course, checking waist circumference is an easy and less expensive method that can help enrich the results of this research. Therefore, it is suggested to check and add waist circumference and its relationship with BMI.

Response: Thank you very much for your feedback and suggestions. Regarding the use of CGSS data, it is indeed true that we do not have waist circumference information available in the dataset for analysis. Therefore, we are unable to include waist circumference data and its relationship with BMI in this study. In our research, we have focused on analyzing the available information in the CGSS data to explore the relationship between BMI and other factors.

2. It is suggested to compare BMI in different regions of China (urban and rural).

Response: Thank you for your valuable feedback. We have already conducted a comparison of BMI in different regions of China (urban and rural) in the manuscript. The relevant text has been emphasized in red for clarity, and the corresponding figures have been included in the supplementary materials. We hope these adjustments address your suggestion.

3. It is suggested that the results of this study be compared with the results in other countries of this region.

Response: Thank you for your suggestion. We have indeed included a comparison of the results of this study with those from other countries in the same region in the discussion section of the manuscript. We have emphasized this comparison using red font to highlight its significance. We hope that this addresses your recommendation.

4. As you know, some factors such as psychosocial status, genetic factors and physical health can affect BMI. Please add explanations about these factors in the discussion.

Response: Thank you for your feedback. We appreciate your suggestion, and we have already incorporated explanations regarding the factors that can affect BMI, such as psychosocial status, genetic factors, and physical health, in the discussion section of the manuscript. The additional content has been marked in red. We hope that these additions enhance the comprehensiveness of our study.

5. Has the relationship between the type of diet (daily calorie intake) and BMI been investigated? Please explain.

Response: Thank you for your inquiry. We did not investigate the relationship between diet type (daily calorie intake) and BMI in our study because we did not have access to relevant data for this specific aspect. However, we have acknowledged the importance of diet as a potential factor influencing BMI and have discussed it in the context of existing research. We have also provided references to other studies that have explored this relationship in more detail. The additional content has been marked in red.

---

## [Editor Report · Decision Letter 2]

27 Dec 2023

Prevalence of Body Mass Index Categories Among Adults living alone in China: Observational Study

PONE-D-23-14813R2

Dear Dr. Yan,

We’re pleased to inform you that your manuscript has been judged scientifically suitable for publication and will be formally accepted for publication once it meets all outstanding technical requirements.

Kind regards,

Zhuo Chen, Ph.D.

Academic Editor

PLOS ONE
---

## [Editor Report · Acceptance letter]

24 Jan 2024

PONE-D-23-14813R2 

PLOS ONE

Dear Dr. Yan, 

I'm pleased to inform you that your manuscript has been deemed suitable for publication in PLOS ONE. Congratulations! Your manuscript is now being handed over to our production team.

Kind regards, 

on behalf of

Prof. Zhuo Chen 

Academic Editor

PLOS ONE